# Effect of the Temperature on the Process of Preferential Solvation of 1,4-Dioxane, 12-Crown-4, 15-Crown-5 and 18-Crown-6 Ethers in the Mixture of *N*-Methylformamide with Water: Composition of the Solvation Shell of the Cyclic Ethers

**DOI:** 10.3390/ijms24108934

**Published:** 2023-05-18

**Authors:** Małgorzata Jóźwiak, Monika A. Trzmielak, Michał Wasiak

**Affiliations:** Department of Physical Chemistry, Faculty of Chemistry, University of Lodz, Pomorska 165, 90-236 Lodz, Poland; monika.trzmielak@wp.pl (M.A.T.); michal.wasiak@chemia.uni.lodz.pl (M.W.)

**Keywords:** 1,4-dioxane, 12-crown-4, 15-crown-5 ether, 18-crown-6 ether, enthalpy of solution, preferential solvation, *N*-methylformamide–water mixtures

## Abstract

The aim of the work was to analyze the preferential solvation process, and determine the composition of the solvation shell of cyclic ethers using the calorimetric method. The heat of solution of 1,4-dioxane, 12-crown-4, 15-crown-5 and 18-crown-6 ethers in the mixture of *N*-methylformamide with water was measured at four temperatures, 293.15 K, 298.15 K, 303.15 K, and 308.15 K, and the standard partial molar heat capacity of cyclic ethers has been discussed. 18-crown-6 (18C6) molecules can form complexes with NMF molecules through the hydrogen bonds between –CH_3_ group of NMF and the oxygen atoms of 18C6. Using the model of preferential solvation, the cyclic ethers were observed to be preferentially solvated by NMF molecules. It has been proved that the molar fraction of NMF in the solvation shell of cyclic ethers is higher than that in the mixed solvent. The exothermic, enthalpic effect of preferential solvation of cyclic ethers increases with increasing ring size and temperature. The increase in the negative effect of the structural properties of the mixed solvent with increase in the ring size in the process of preferential solvation of the cyclic ethers indicates an increasing disturbance of the mixed solvent structure, which is reflected in the influence of the energetic properties of the mixed solvent.

## 1. Introduction

Scientists dealing with organic synthesis are looking for solvents in which the chemical reactions that are carried out will have the highest possible efficiency. Not only are pure solvents taken into account, but also mixed water–organic and organic–organic mixtures that significantly affect the solvation of the reactants. The solvation process plays a very important role in the interactions between molecules of solvent, solvent and solute, as well as solute and solute [1,2,3]. Studying the solvation process, including preferential solvation, is very important from the perspective of the interactions between the molecules of reactants. The process of preferential solvation is defined as a difference in the mixed solvent composition between the bulk solvent and the solvation shell [4,5]. Moreover, if the molecules exhibit a hydrophobic character and the reaction is carried out in an aqueous environment, hydrophobic hydration should be taken into account.

Studies on preferential solvation have been carried out for a long time, mostly using solubility measurements [6,7,8]. Marcus extensively investigated this phenomenon, utilizing the solubility of investigated substances in water [9], organic solvents [10] and mixed solvents [11]. The aim of these studies was to observe whether the molecules undergo self-association, preferentially interact with the co-solvent, or whether the solute molecules are preferentially solvated by the molecules of the one of the components of mixed solvents. This is very important from the point of view of, e.g., pharmacology and the effect of the drug, or the occurrence of chemical reactions, or a change in the mechanism of the reaction due to change in the solvent. Similar studies were conducted using a spectroscopic method [12]. There are only a few publications on this topic that utilize the calorimetric method. Waghorn proposed using the calorimetric method to calculate the composition of the solvation shell [13,14]; however, this does not always lead to satisfactory results. Therefore, there have been attempts to utilize another theory to analyze the preferential solvation process. In the present publication, which is a continuation of our research [15,16], we used the model of preferential solvation proposed by Covington and modified by Balk and Somsen [17,18,19] for the analysis of the calorimetric data, calculating the solvation shell of the solute in this process. For this purpose, we chose the cyclic ethers (1,4-dioxane, 12-crown-4 (12C4), 15-crown-5 (15C5) and 18-crown-6 (18C6)) and *N*-methylformamide–water (NMF + W) mixed solvent.

Crown ethers are an interesting class of compounds, characterised by a hydrophobic–hydrophilic character. They can form selective complexes with cations and small organic compounds [20,21]. For this reason, they are widely used in chemistry, biology and medicine [22,23,24,25,26,27,28]. Due to their properties, crown ethers can be hydrophobically hydrated in water [29,30] as well as preferentially solvated in mixed solvents [31].

*N*-methylformamide is a solvent with hydrophilic properties and can be treated as a derivative of formamide (F), obtained by replacing one of the hydrogen atoms connected to the nitrogen atom of formamide with a –CH_3_ group. The obtained results of the preferential solvation study in the NMF + W mixture can be compared with the analogous ones referring to the F + W mixture.

## 2. Results and Discussion

### 2.1. Enthalpy of Solution and Heat Capacity

Figure 1 shows the curves in the standard solution enthalpy of investigated cyclic ethers at four temperatures within the range 293.15–308.15, with K as a function of the mole fraction of water, *x*_w_ in the NMF + W mixture.

As can be seen in Figure 1, the enthalpy of dissolution is slightly temperature-dependent. The temperature dependence increases with an increase in the size of the cyclic ether ring. The shapes of all the curves of $\Delta_{\mathrm{sol}}H_{}^{o}=f(x_{W})$ are similar for the investigated cyclic ethers, except 1,4-dioxane.

The course of curves is monotonic and decreases with an increase of *x*_W_ in the mixture. In the case of 1,4-dioxane, the values of the function $\Delta_{\mathrm{sol}}H_{}^{o}=f(x_{W})$ increase with an increase in the water content in the mixture of up to *x*_W_ ≈ 0.5 and then decrease. This is probably related to the structure of the molecules of cyclic ethers. The 12C4, 15C5, and 18C6 are included in crown ethers with a hydrophilic character inside the molecule and a hydrophobic character outside the molecules. The molecules of 1,4-dioxane are smaller than other cyclic ethers, and it is not possible to unambiguously locate the hydrophobic and hydrophilic nature of this part of the molecule.

The data of the standard molar enthalpy of solution of the investigated cyclic ethers allow for us to calculate the standard molar heat capacity of the solution of cyclic ethers using Equation (1).
(1)$$\Delta_{\mathrm{sol}}C_{p}^{o}={\left(\partial \Delta_{\mathrm{sol}}H^{o}/\partial T\right)}_{p}$$

Then, the standard partial molar heat capacity ($C_{p,2}^{o}$) of cyclic ethers can be calculated using Equation (2):(2)$$C_{p,2}^{o}=\Delta_{\mathrm{sol}}C_{p}^{o}+C_{p}^{*}$$
where $C_{p}^{*}$ is the molar heat capacity of pure cyclic ether taken from the literature [32,33].

The results obtained at 298.15 K are presented in Figure 2.

As seen in Figure 2, the values of the function $C_{p,2}^{o}=f(x_{W})$ increase with an increase in the cyclic ethers’ size. It is possible to see that the shapes of this function are similar for all cyclic ethers in the high water content in the NMF + W mixture (*x*_W_ > 0.85). In this range, the increase in the function $C_{p,2}^{o}=f(x_{W})$ is observed with an increase in the water content in the mixture. This is most likely related to the hydrophobic hydration of cyclic ethers. The analogous observation was discussed by Somsen [34] and Castagnolo [35] regarding the heat capacity of tetra-n-butyammonium bromide, and by Desnoyers et al. [36] in the case of acetamide, dimethylsulfoxide, and acetone in mixed water–organic solvents. The shape of curves was found to be very similar to that obtained in our investigation, and is related to the hydrophobic hydration of solute molecules. In the second range of water content (*x*_W_ < 0.85), the values of the function $C_{p,2}^{o}=f(x_{W})$ do not change with the water content in the 1,4-dioxane, 12C4 and 15C5 mixture. This means that even if there are any changes in the structure of the solution, they are small and do not influence the shape of $C_{p,2}^{o}=f(x_{W})$ function in this composition range. In the case of 18C6, the value of this function decreases with an increase in water content of up to *x*_W_ ≈ 0.85. This means that, in this system, there are some changes in the interactions due to the dissolution of 18C6 molecules in the NMF + W mixture. The answer to this question is not as obvious. In the literature, we found no information on the interactions between 18C6 molecules and NMF, as was the case for interactions between 18C6 molecules and formamide (F). 18C6 molecules were found to form complexes with the F molecules in the solid state [37,38]. In our previous paper, we suggested that this kind of complex also exists in solution, and this is reflected by the slight increase in the heat capacity in the low water content range in the mixture with formamide (*x*_W_ < 0.1) [16].

In the NMF + W mixture, a change in the function $C_{p,2}^{o}=f(x_{W})$ is observed in a wider mixture composition range than that found in the F + W mixture. This is likely related to the structure of the NMF + W and F + W mixtures. The NMF molecules form a weaker network of hydrogen bonds than F [39]. The energy of hydrogen bonds in NMF is slightly lower than in F [40]. When formamide is added to water, its original structure is destroyed, creating a new stable system [41,42]. In the case of pure NMF, molecules interact with each other through linear hydrogen bonds. In pure water, the molecules are also associated through hydrogen bonds. The addition of NMF to water causes the formation of new hydrogen bonds between water molecules and NMF (C=O∙∙∙∙H–O), which are not very different from those found in water. This is indicated by the heat capacity [43], viscosity or spectroscopic tests [44]. In addition, Assarsson and Eirich suggested that two water molecules should interact more strongly with the two lone pairs of electrons on the carbonyl oxygen, while the third molecule might interact less with the electron pair on the nitrogen atom. Therefore, it can be presumed that *N*-methylforamide can form three hydrogen bonds with water, whereas this type of interaction is less important in the F + W mixtures [45]. Furthermore, Marcus, while analyzing the preferential solvation parameter δ*x*, noticed that the mixture of NMF + W behaves almost like an ideal solution [9].

In general, the structure of the mixed solvent NMF + W is weaker than that of F + W; therefore, there is the possibility of a specific interaction between 18C6 molecules and NMF molecules in a wider range of the mixture composition than in the case of the F + W mixture. On this basis, it can be assumed that 18C6 molecules form complexes with NMF molecules through hydrogen bonds. We can suppose that 18C6 molecules form complexes by hydrogen bonds between hydrogen atoms of the NMF molecule and an electron pair of oxygen atoms of the 18C6 molecule, as well as between hydrogen atoms of the –CH_3_ group and an electron pair of oxygen atoms of the 18C6 molecule, similarly to the hydrogen bonds of acetonitrile with oxygen atoms of the 18C8 molecule [46], particularly in medium- and low-water content in the mixture. In the F + W mixture, this type of interaction cannot occur.

### 2.2. The Preferential Solvation of Cyclic Ethers

The cyclic ethers, because of their hydrophobic character, are hydrophobically hydrated in mixed solvent with a high water content. In addition, the molecules may be preferentially solvated by the organic component of the mixture.

In order to calculate the enthalpy effect of preferential solvation, we used the model proposed by Covington [17,18] and modified by Balk and Somsen [19]. In this model, the enthalpy of dissolution can be written as the sum of the enthalpy effects associated with hydrophobic hydration and the enthalpy of other effects, e.g., preferential solvation (Equation (3)).
(3)$$\Delta_{\mathrm{sol}}H^{o}(W+Y)=x_{w}\Delta_{\mathrm{sol}}H^{o}\left(W\right)+x_{y}\Delta_{\mathrm{sol}}H^{o}\left(Y\right)+(x_{w}^{n}-x_{w})Hb\left(W\right)+\Delta H^{*}(W+Y)$$
where $\Delta_{\mathrm{sol}}H^{o}(W+Y)$, $\Delta_{\mathrm{sol}}H^{o}\left(W\right)$, $\Delta_{\mathrm{sol}}H^{o}\left(Y\right)$—the standard solution enthalpy of hydrophobic substance in the mixed solvent, in pure water, and in pure organic solvent, respectively;

$x_{y}=(1-x_{w})$—the molar fraction of organic co-solvent in the mixed solvent;

*Hb*(W) − the enthalpic effect of hydrophobic hydration of the solute in pure water

$(x_{w}^{n}-x_{w})Hb\left(W\right)$—the factor connected with hydrophobic hydration in the mixture;

$\Delta H^{*}(W+Y)$—the enthalpic effect of interactions in a solution differing from the hydrophobic hydration of the solute molecules.

By rearranging Equation (3), we can calculate the enthalpic effect other than hydrophobic hydration: $\Delta H^{*}(W+Y)$ (Equation (4)).
(4)$$\Delta H^{*}(W+Y)=\Delta_{\mathrm{sol}}H^{o}\left(W\right)-\{x_{w}\Delta_{\mathrm{sol}}H^{o}\left(W\right)+x_{y}\Delta_{\mathrm{sol}}H^{o}\left(Y)\right\}+(x_{w}^{n}-x_{w})Hb\left(W\right)$$

The method of calculating the enthalpic effect of hydrophobic hydration was described in our previous publication [47]. The values of parameters *n* and *Hb*(W), which are necessary for the calculations, were taken from our publication [30].

In the case of investigated cyclic ethers within the whole mixed-solvent composition range, the value of function $\Delta H^{*}(W+Y)=f(x_{w})$ is negative (Figure 3). Therefore, we can conclude that molecules of cyclic ethers are preferentially solvated by molecules of one component of the NMF + W mixture and $\Delta H^{*}(W+Y)\equiv \Delta_{\mathrm{PS}}H^{E}(W+Y)$.

As shown in Figure 3 the exothermic, enthalpic effect of preferential solvation increases with an increase of cyclic ethers’ size and temperature. Comparing this effect for cyclic ethers it can be seen that, in the NMF + W mixture, it is less negative than in the F + W mixture [16]. As the temperature increases, the interactions between the molecules of the mixed solvent start to weaken and it will be easier for one of the components of the mixture to interact with the molecules of the cyclic ethers.

For a further analysis of the obtained results, we used the theory of preferential solvation [17,18], hoping that this would answer the question of which solvent molecules are preferentially solvated by cyclic ether molecules. In this theory, the process of preferential solvation of solute (S) can be described by Reaction (5).
S(W*_i_*_−1_NMF*_r_*_+1−*i*_) + W ⇆ S(W*_i_*NMF*_r_*_−*i*_) + NMF(5)
where (1 ≤ *i* ≤ *r*).

The equilibrium constant *K*_i_ described the process of changes in the structure of the solute solvation sphere (S) caused by the changes in the mixed solvent composition because of preferential solvation. This can be represented by Equation (6).
(6)$$K_{i}=K^{1/r}[(r+1-i)/i]$$
where *K*—the overall constant of equilibrium of all processes, $K=\overset{r}{\underset{i=1}{\Pi }}K_{i}$, $r=r_{\mathrm{NMF}}+r_{W}$
where $r_{\mathrm{NMF}}$ and $r_{W}$ are the number of molecules of NMF and W in the solvation shell of solute (S), respectively.

The enthalpic effect of the preferential solvation of the process presented by Equations (5) and (6) is described by Equation (7).
(7)$$\Delta_{\mathrm{PS}}H^{E}(\mathrm{NMF}\;+\;W)=rRT\left[\left\{\frac{1-x_{W}}{\left(1-x_{W}\right)+K^{1/r}\cdot x_{W}}\right\}-\left(1-x_{W}\right)\right]\cdot \ln K^{1/r}$$

Using Equation (7) and non-linear regression, parameters *r* and *K*^1/*r*^ were calculated and are presented in Table 1.

In Table 1, parameter *K*^1/*r*^ is shown to be lower than 1, which means that the functional groups in cyclic ethers are preferentially solvated by the organic component (NMF) of the mixture [19]. Moreover, the values of parameter *r* increase with increases in the ring size of cyclic ethers and decrease with increasing temperature.

The equilibrium with the constant *K*, described by Equation (5), represents the reverse process, i.e., the solvation by water molecules. Thus, in the case of preferential solvation by NMF molecules, the equilibrium constant of this process will be equal to *K*’ = 1/*K* (Table 1). As shown in Table 1, the equilibrium constants of preferential solvation increase with increases inring size of cyclic ethers and decrease with increasing temperature. An increase in temperature causes a weakening of the interactions between molecules of water and NMF, as well as between molecules of cyclic ethers and NMF due to an increase in thermal motion.

Specific interactions between cyclic ether molecules and NMF molecules may occur through the hydrogen bonds between hydrogen atoms (one at the nitrogen atom and the other in the carbonyl group) and oxygen atoms in the cyclic ether molecules, especially in the areas of medium- and low-water content in the mixture. In addition, molecules of 18C6 can form complexes with NMF molecules by interacting with the –CH_3_ group, which we wrote about in Section 2.1.

The thermodynamic function (Gibbs’ energy, enthalpy and entropy) of transferring substance (S) from W to the mixed solvent NMF + W in the preferential solvation process can be described by Equations (8)–(10), proposed by Balk and Somsen [19].
(8)$$\Delta_{\mathrm{tr}}G(W\to \mathrm{NMF}\;+\;W)=-rRT\ln[K^{-1/r}x_{\mathrm{NMF}}+x_{W}]$$
(9)$$\Delta_{\mathrm{tr}}H\left(W\to \mathrm{NMF}\;+\;W\right)=rRT\left[\frac{1-x_{W}}{\left(1-x_{W}\right)+K^{1/r}x_{W}}\right]\cdot \ln K^{1/r}$$
(10)$$T\Delta_{\mathrm{tr}}S{\left(W\to \mathrm{NMF}\;+\;W\right)}_{\left(A\right)}=-r_{W}RT\ln\left(\frac{r_{W}}{rx_{W}}\right)-r_{\mathrm{NMF}}RT\ln\left(\frac{r_{\mathrm{NMF}}}{r\left(1-x_{W}\right)}\right)$$
(11)$$T\Delta_{\mathrm{tr}}S{(W\to \mathrm{NMF}\;+\;W)}_{\left(B\right)}=\Delta_{\mathrm{tr}}H(W\to \mathrm{NMF}\;+\;W)-\Delta_{\mathrm{tr}}G(W\to \mathrm{NMF}\;+\;W)$$

Using Equation (11) the values of $T\Delta_{\mathrm{tr}}S{\left(W\to \mathrm{NMF}\;+\;W\right)}_{\left(B\right)}$ were calculated. Then, the values of the number of water molecules in the solvation sphere, *r*_W_, were calculated using Equation (10) by choosing the *r*_W_ values so that equality was met: $T\Delta_{\mathrm{tr}}S{\left(W\to \mathrm{NMF}\;+\;W\right)}_{\left(B\right)}$ = $T\Delta_{\mathrm{tr}}S{\left(W\to \mathrm{NMF}\;+\;W\right)}_{\left(A\right)}$. Next, the values of the number of molecules of *N*-methylformamide in the solvation sphere, *r*_NMF_, and the mole fraction of NMF, *y*_NMF_, in the solvation sphere were calculated. Appendix A, showing detailed calculations, are presented in the Appendix A. As shown in these tables, the mole fraction of NMF in the solvation sphere of cyclic ethers, *y*_NMF_, is almost independent of temperature, so the mean value was calculated for each cyclic ethers and is presented in Table 2. In Table 2, the mole fraction of water, *x*_W_, and NMF, *x*_NMF_ in the NMF + W mixture are also presented. 

In Figure 4, the relationship $y_{\mathrm{NMF}}=f\left(x_{\mathrm{NMF}}\right)$ is presented. As seen in Table 2 and Figure 4, the mole fractions of *N*-methylformamide in the solvation sphere (*y*_NMF_) are higher than those in the mole fraction of *N*-methylformamide in the NMF + W (*x*_NMF_) mixture. Thus, it is clear that cyclic ethers are preferentially solvated by *N*-methylformamide molecules.

For 1,4-dioxane, the values of the mole fraction of NMF in the solvation sphere are higher than those in the mole fraction of F in this sphere [16]; however, in the case of other cyclic ethers, the opposite situation is observed [16]. This is most likely due to the different sizes of the NMF and F molecules, as well as other interactions between the NMF and F molecules and the cyclic ether molecules.

### 2.3. The Effect of Structural and Energetic Properties of NMF + W Mixture on the Preferential Solvation

In our previous paper [31], we proposed describing the effect of the structural and energetic properties of the mixed solvent on the enthalpic effect of preferential solvation ($\Delta_{\mathrm{PS}}H^{E}(\mathrm{NMF}\;+\;W)$) (Equation (12)).
(12)$$\Delta_{\mathrm{PS}}H^{E}(\mathrm{NMF}\;+\;W)=bV^{E}\left(\mathrm{NMF}+W\right)+cH^{E}\left(\mathrm{NMF}+W\right)$$

The excess molar volume ($V^{E}$) expresses the structural properties of the mixed solvent, i.e., direct interactions between, e.g., the hydrogen bond or structuredness of the mixture. The excess enthalpy ($H^{E}$) represents the energetic properties of the mixture, which are connected with the structure of the mixed solvent. Using Equation (12) and the values of the excess molar volume [48] as well as the excess molar enthalpy [49] for the NMF + W mixture obtained from literature, the factors *bV*^E^ and *cH*^E^, have been calculated and are presented in Figure 5.

As shown in Figure 5, the contribution of structural properties decreases, but the energetic properties of mixture the NMF + W increase with the increasing ring size of the cyclic ethers. This suggests that the cyclic ethers strengthen the structure of the NMF + W mixture; the larger the cyclic ether ring, the stronger the structure. The increasing energetic effect indicates the increasing difficulty of incorporating cyclic ether molecules into the existing mixture. This is also reflected in Figure 4, where it can be seen that the molar fraction of NMF in the solvation shell of cyclic ethers decreases with the increase in the cyclic ring.

The negative contribution of structural properties and the positive contribution of the energetic properties of mixture of hexamethylphosphortriamide (HMPA) with water is observed in the case of the enthalpic effect, which differs to the hydrophobic hydration of 15C5 [16]. HMPA is a solvent with very strong hydrophobic character, which strengthens the structure of water.

In our case, NMF interacts with water through hydrogen bonds and forms a new, stable structure with water, which is also reflected by the negative contribution of the structural properties and the positive contribution of the energetic properties of the mixed solvent to the solvation of the cyclic ethers.

## 3. Experimental Section

### 3.1. Materials

Suppliers, purity, a method of purification and the water content of the compounds used for measurements (urea, potassium chloride, 1,4-dioxane, 12-crown-4, 15-crown-5, 18-crown-6 and *N*-methylformamide) are shown in Table 3. To prepare the aqueous solutions, doubly distilled water was used.

### 3.2. Methods

The heat of solution measurements was found for 1,4-dioxane, 12-crown-4, 15C5 and 18C6 using an “isoperibol” type calorimeter, as described in the literature [50] within the whole mole fraction range at [(293.15, 298.15, 303.15, 308.15) ± 0.01] K. To validate the calorimeter, the standard enthalpy of the solution of urea and potassium chloride (KCl) (calorimetric standard US, NBS) was used [51,52]. The mean value of the enthalpy of urea solution in water, obtained by us at 298.15 K from seven independent measurements, was (15.31 ± 0.06) kJ∙mol^−1^ (literature data 15.31∙kJ mol^−1^ [53], 15.28 kJ∙mol^−1^ [54]. That for KCl in water was (17.55 ± 0.05) kJ∙mol^−1^ (literature data 17.58 kJ∙mol^−1^ [51,52].

We have not observed the concentration dependence of the enthalpy of dissolution of cyclic ethers in our study. Therefore, the values obtained from six to eight independent measurements of the dissolution heat were used to calculate the average value of the standard enthalpy of dissolution, $\Delta_{\mathrm{sol}}H_{}^{o}$ (Table 4, Table 5, Table 6 and Table 7).

## 4. Conclusions

The investigation of the dissolution enthalpy of cyclic ethers (1,4-dioxane, 12C4, 15C5 and 18C6) in the mixture of *N*-methylformamide and water (NMF + W) at four temperatures led to the following conclusion regarding the solvation of the cyclic ethers.

The exothermic enthalpic effect of the solvation process increases with the increase in the size of the cyclic ethers ring.Molecules of 1,4-dioxane, 12C4, 15C5, and 18C6 are preferentially solvated by the NMF molecules in a mixture of *N*-methylformamide and water.The exothermic, enthalpic effect of preferential solvation increases with increasing temperature for the cyclic ethers (1,4-dioxane, 12C4, 15C5 and 18C6).The total number of NMF and W molecules in the solvation sphere of the molecules of cyclic ethers increases with increasing ether ring size.The mole fraction of NMF in the solvation sphere of cyclic ethers is higher than that of the molecules in the NMF + W mixture.The increase in the negative effect of the mixed solvent’s structural properties with the increase in the size of the cyclic ring during the process of preferential solvation of the cyclic ethers indicates an increasing disturbance to the mixed solvent structure, which is reflected in the influence of the energetic properties of the mixed solvent.

## Figures and Tables

**Figure 1 ijms-24-08934-f001:**
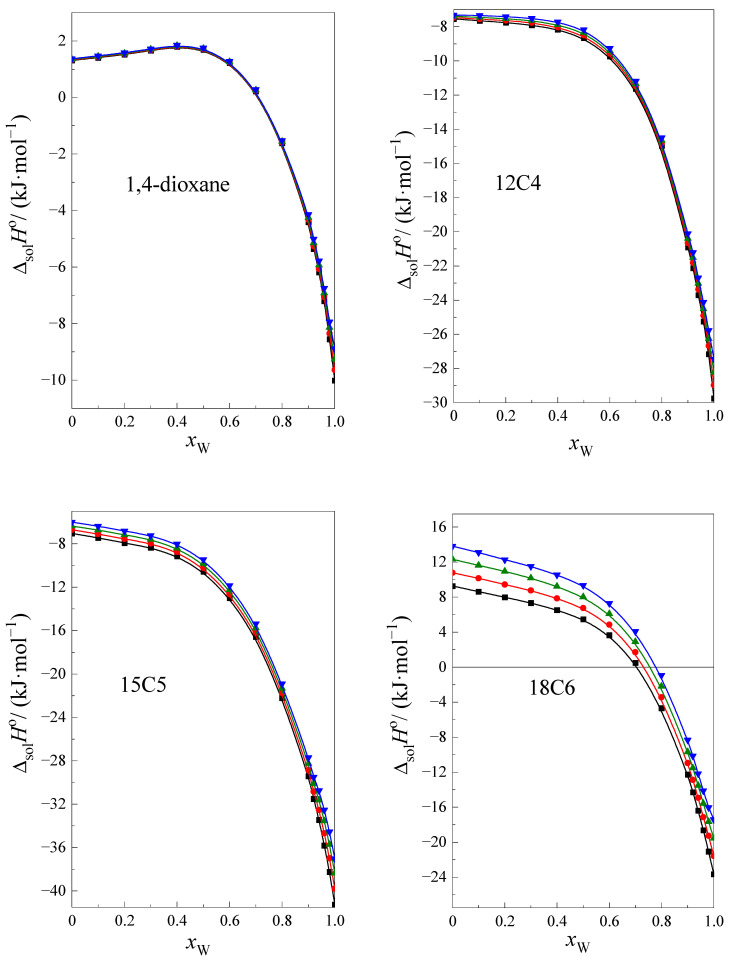
The standard solution enthalpy of 1,4-dioxane, 12C4, 15C5 and 18C6 in the NMF + W mixtures as a function of the molar fraction of water *x*_W_ at ■, 293.15 K; ●, 298.15 K; ▲, 303.15 K; ▼, 308.15 K.

**Figure 2 ijms-24-08934-f002:**
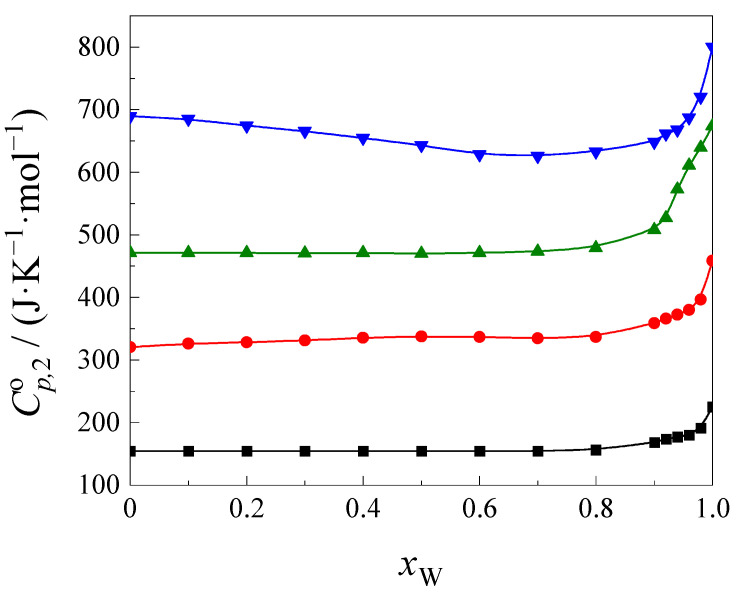
The standard partial molar heat capacity of cyclic ethers: ■, 1,4-dioxane, ●, 12C4, ▲, 15C5, ▼, 18C6, as a function of *x*w in the NMF + W mixture at 298.15 K.

**Figure 3 ijms-24-08934-f003:**
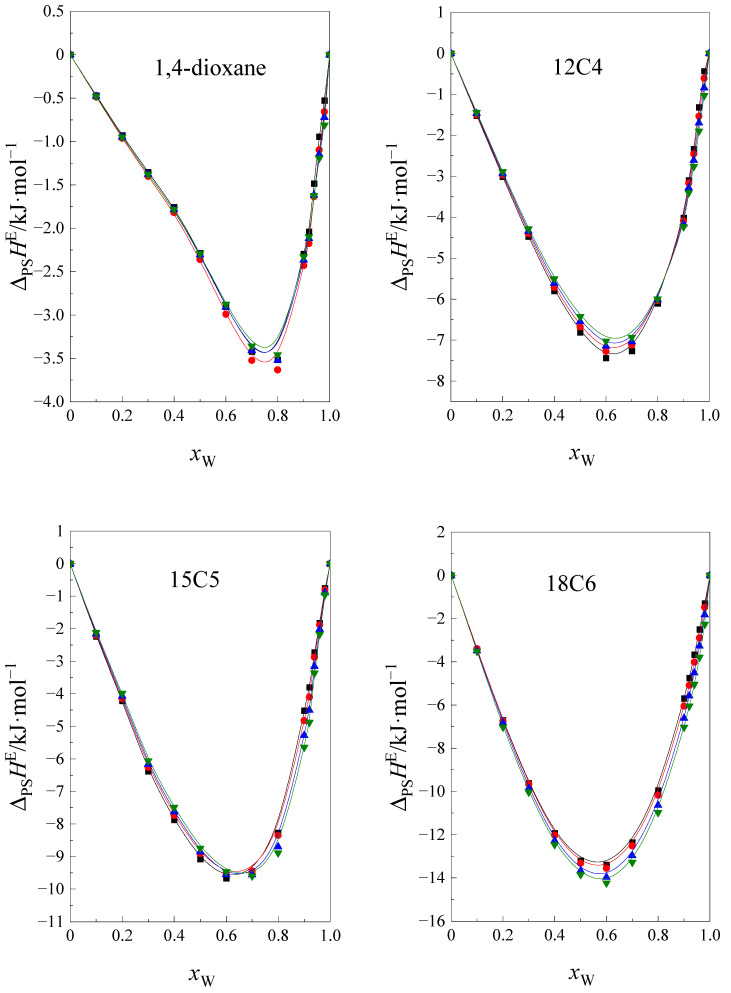
The enthalpic effect of the preferential solvation process $\Delta_{\mathrm{PS}}H^{E}(W+Y)$ of cyclic ethers 1,4 dioxane, ■, 12C4, ●, 15C5, ▲, 18C6, ▼ in a function of the mole fraction of water ($x_{W}$) in the NMF + W mixture at 293.15 K, 298.15 K, 303.15 K, and 308.15 K.

**Figure 4 ijms-24-08934-f004:**
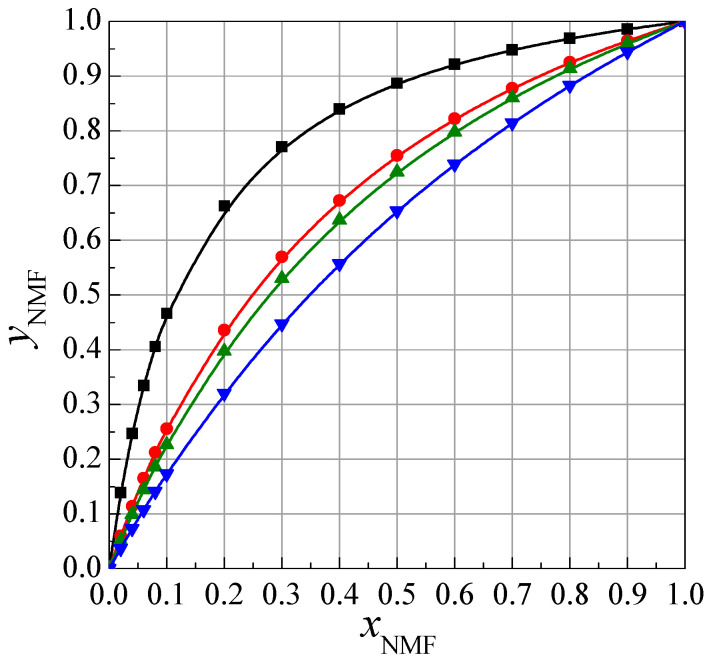
Dependence of the mole fraction of NMF in the solvation sphere ($y_{\mathrm{NMF}}$) of cyclic ethers: ■, 1,4-dioxane; ●, 12C4; ▲, 15C5; ▼, 18C6, as a function of the mole fraction of NMF in the NMF + W mixture.

**Figure 5 ijms-24-08934-f005:**
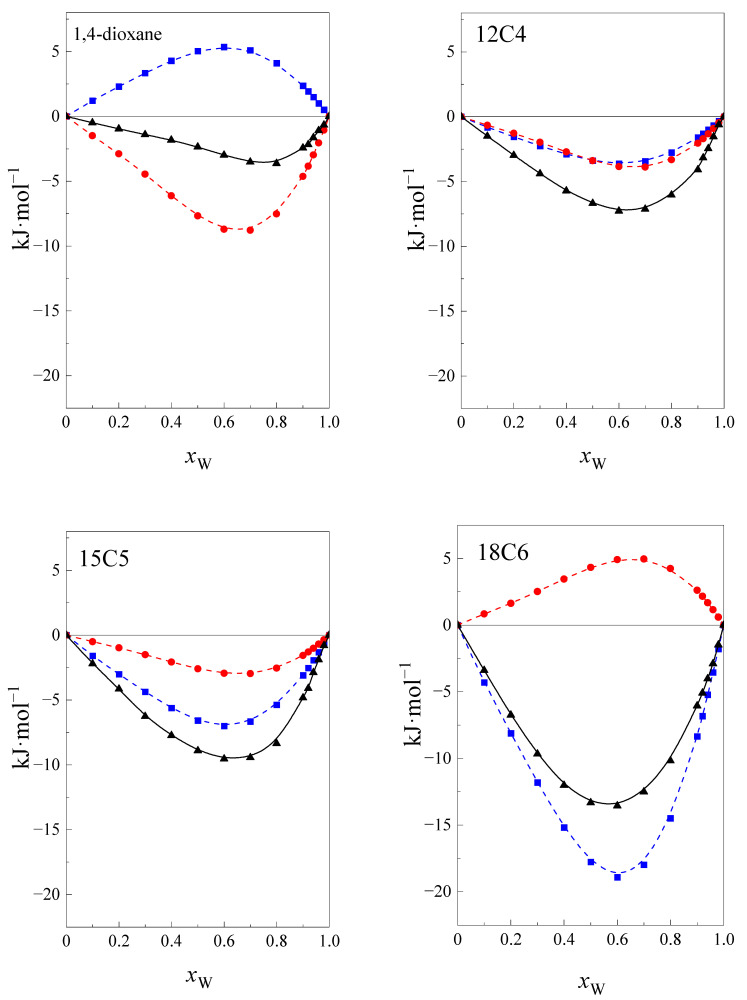
The contribution of structural $bV^{E}(\mathrm{NMF}\;+\;W)$, ■, and energetic $cH^{E}(\mathrm{NMF}\;+\;W)$, ●, properties of the NMF + F mixture in the preferential solvation of cyclic ethers $\Delta_{\mathrm{PS}}H^{E}(\mathrm{NMF}\;+\;W)$, ▲.

**Table 1 ijms-24-08934-t001:** Parameters of Equation (7) for the preferential solvation of cyclic ethers in the NMF + W mixtures.

Cyclic Ethers	*T*/K	*r*	*K* ^1/*r*^	*K*	*K*’ = 1/*K*	*R* ^2^
1,4-dioxane	293.15	1.37 ± 0.23	0.14 ± 0.02 ^a^	6.76 × 10^−2^	14.78	0.948280
	298.15	1.30 ± 0.19	0.13 ± 0.02	7.05 × 10^−2^	14.19	0.95662
	303.15	1.22 ± 0.16	0.12 ± 0.02	7.53 × 10^−2^	13.29	0.96237
	308.15	1.17 ± 0.14	0.1 ± 0.01	8.37 × 10^−2^	11.95	0.96703
12C4	293.15	11.50 ± 1.81	0.36 ± 0.03	7.90 × 10^−6^	12.6625 × 10^4^	0.98867
	298.15	9.93 ± 1.11	0.34 ± 0.02	2.23 × 10^−5^	4.4917 × 10^4^	0.99330
	303.15	8.46 ± 0.62	0.31 ± 0.01	4.98 × 10^−5^	2.0094 × 10^4^	0.99652
	308.15	7.19 ± 0.33	0.29 ± 0.01	1.36 × 10^−4^	0.7334 × 10^4^	0.99831
15C5	293.15	20.72 ± 3.48	0.42 ± 0.03	1.56 × 10^−8^	6.4 × 10^7^	0.99085
	298.15	16.66 ± 2.21	0.38 ± 0.02	1.10 × 10^−7^	9.1 × 10^7^	0.99273
	303.15	12.86 ± 1.43	0.33 ± 0.02	6.43 × 10^−7^	1.6 × 10^7^	0.99306
	308.15	10.42 ± 1.03	0.30 ± 0.02	3.56 × 10^−6^	2.8 × 10^6^	0.99306
18C6	293.15	68.26 ± 8.87	0.57 ± 0.02	2.17 × 10^−17^	4.6 × 10^16^	0.99780
	298.15	53.84 ± 4.86	0.53 ± 0.02	1.43 × 10^−15^	7.0 × 10^14^	0.99858
	303.15	41.66 ± 3.27	0.48 ± 0.01	5.25 × 10^−14^	1.9 × 10^13^	0.99851
	308.15	33.72 ± 3.53	0.44 ± 0.02	9.49 × 10^−13^	1.1 × 10^12^	0.99648

± is the standard deviation. *R*^2^ is the regression coefficient.

**Table 2 ijms-24-08934-t002:** The mole fraction of NMF in the solvation sphere of the cyclic ethers (*y*_NMF_), depending on the mole fraction of water (*x*_W_) or NMF (*x*_NMF_) in the mixture NMF + W.

		1,4-Dioxane	12-Crown-4	15-Crown-5	18-Crown-6
$x_{W}$ ^a^	$x_{\mathrm{NMF}}$ ^b^	$y_{\mathrm{NMF}}$ ^c^	$y_{\mathrm{NMF}}$ ^c^	$y_{\mathrm{NMF}}$ ^c^	$y_{\mathrm{NMF}}$ ^c^
0.000	1.000	1.000	1.000	1.000	1.000
0.100	0.900	0.986 ± 0.001	0.965 ± 0.003	0.962 ± 0.006	0.947 ± 0.006
0.200	0.800	0.969 ± 0.002	0.925 ± 0.007	0.919 ± 0.012	0.888 ± 0.011
0.300	0.700	0.948 ± 0.004	0.878 ± 0.010	0.868 ± 0.018	0.822 ± 0.016
0.400	0.600	0.922 ± 0.005	0.822 ± 0.014	0.809 ± 0.025	0.749 ± 0.021
0.500	0.500	0.887 ± 0.008	0.755 ± 0.018	0.739 ± 0.031	0.665 ± 0.025
0.600	0.400	0.840 ± 0.010	0.673 ± 0.021	0.654 ± 0.036	0.570 ± 0.028
0.700	0.300	0.771 ± 0.013	0.569 ± 0.024	0.549 ± 0.040	0.460 ± 0.028
0.800	0.200	0.663 ± 0.017	0.436 ± 0.024	0.416 ± 0.039	0.333 ± 0.025
0.900	0.100	0.466 ± 0.018	0.256 ± 0.018	0.241 ± 0.030	0.182 ± 0.017
0.920	0.080	0.406 ± 0.018	0.212 ± 0.016	0.199 ± 0.026	0.148 ± 0.014
0.940	0.060	0.334 ± 0.016	0.165 ± 0.013	0.155 ± 0.021	0.113 ± 0.011
0.960	0.040	0.247 ± 0.014	0.114 ± 0.010	0.107 ± 0.015	0.077 ± 0.008
0.980	0.020	0.138 ± 0.009	0.059 ± 0.005	0.055 ± 0.009	0.039 ± 0.004
1.000	0.000	0.000	0.000	0.000	0.000

^a^ is the mole fraction of water in the NMF + W mixture. ^b^ is the mole fraction of NMF in the NMF + W mixture. ^c^ is the mole fraction of NMF in the solvation sphere of cyclic ethers in the NMF + W mixture. ± is the standard deviation.

**Table 3 ijms-24-08934-t003:** Materials.

Chemical Name	Source	Mole Fraction Purity ^a^	Purification Method	Mass Fraction of Water ^b^
Urea(U)	Sigma-Aldrich, (Poznan, Poland)	>0.995	Recrystallisation from ethanol and drying under reduced pressure to constant mass.	−
Potassium chloride (KCl)	Sigma-Aldrich, (Poznan, Poland)	>0.99	Drying under reduced pressure to constant mass.	−
1,4-dioxane	Sigma-Aldrich, (Poznan, Poland)	>99%	Drying under reduced pressure.	2 × 10^−4 b^
12-crown-4 ethers	Sigma-Aldrich, (Poznan, Poland)	98%		7 × 10^−4 b^
15-crown-5 (15C5)	Sigma-Aldrich, (Poznan, Poland)	0.98		1 × 10^−3^
18-crown-6 (18C6)	Sigma-Aldrich, (Poznan, Poland)	≥0.99	Recrystallisation from hexane and drying under reduced pressure.	−
*N*-methylformamide (NMF)	Aldrich	0.99	Drying using 4A molecular sieves and calcium oxide and distillation under reduced pressure.	3 × 10^−4^

^a^ Declared by the supplier. ^b^ Determined by Karl Fischer method.

**Table 4 ijms-24-08934-t004:** Standard molar enthalpy of solution ($\Delta_{\mathrm{sol}}H_{}^{o}$) and molality (*m*) of 1,4-dioxane in the NMF + W mixtures at *T* = (293.15, 298.15, 303.15, 308.15) K.

	*T* = 293.15 K		*T* = 298.15 K		*T* = 303.15 K		*T* = 308.15 K	
$x_{W}$ **^a^**	$\frac{m^{b}\cdot 10_{}^{3}}{\mathrm{mol}\cdot {\mathrm{kg}}^{-1}}$	$\frac{\Delta_{\mathrm{sol}}H_{}^{o}}{\mathrm{kJ}\cdot {\mathrm{mol}}^{-1}}$	$\frac{m^{b}\cdot 10_{}^{3}}{\mathrm{mol}\cdot {\mathrm{kg}}^{-1}}$	$\frac{\Delta_{\mathrm{sol}}H_{}^{o}}{\mathrm{kJ}\cdot {\mathrm{mol}}^{-1}}$	$\frac{m^{b}\cdot 10_{}^{3}}{\mathrm{mol}\cdot {\mathrm{kg}}^{-1}}$	$\frac{\Delta_{\mathrm{sol}}H_{}^{o}}{\mathrm{kJ}\cdot {\mathrm{mol}}^{-1}}$	$\frac{m^{b}\cdot 10_{}^{3}}{\mathrm{mol}\cdot {\mathrm{kg}}^{-1}}$	$\frac{\Delta_{\mathrm{sol}}H_{}^{o}}{\mathrm{kJ}\cdot {\mathrm{mol}}^{-1}}$
0.00	17.05–18.96	1.31 ± 0.05 ^c^	8.44–8.52	1.33 ± 0.06	8.72–9.73	1.35 ± 0.06	4.23–4.88	1.37 ± 0.05
0.10	22.88–28.07	1.41 ± 0.06	6.88–6.92	1.43 ± 0.06	9.88–9.94	1.45 ± 0.08	4.88–4.93	1.47 ± 0.05
0.20	22.11–22.13	1.52 ± 0.05	6.53–6.81	1.54 ± 0.04	8.98–9.09	1.56 ± 0.04	9.74–9.81	1.58 ± 0.06
0.30	17.80–20.31	1.66 ± 0.07	6.34–6.43	1.68 ± 0.05	4.67–4.98	1.70 ± 0.08	9.31–9.48	1.72 ± 0.06
0.40	14.99–16.42	1.79 ± 0.06	7.26–7.72	1.81 ± 0.08	9.31–9.54	1.83 ± 0.04	9.10–9.44	1.85 ± 0.07
0.50	13.38–14.14	1.69 ± 0.04	7.31–7.73	1.71 ± 0.05	7.68–7.88	1.73 ± 0.05	7.57–7.75	1.75 ± 0.05
0.60	13.70–14.08	1.22 ± 0.04	8.22–8.45	1.24 ± 0.07	6.84–8.35	1.26 ± 0.05	7.63–9.77	1.28 ± 0.05
0.70	12.01–13.47	0.22 ± 0.04	8.11–8.29	0.24 ± 0.05	6.55–9.37	0.26 ± 0.04	7.73–8.63	0.28 ± 0.06
0.80	5.14–6.50	−1.61 ± 0.06	7.89–7.97	–1.58 ± 0.08	5.39–5.83	−1.56 ± 0.04	8.00–8.91	−1.52 ± 0.06
0.90	13.96–14.33	−4.41 ± 0.06	7.20–7.37	–4.33 ± 0.04	5.53–7.61	−4.25 ± 0.06	7.84–7.86	−4.14 ± 0.08
0.92	3.64–6.22	−5.36 ± 0.04	8.19–8.22	–5.26 ± 0.08	5.48–6.27	−5.14 ± 0.06	7.73–8.01	−5.01 ± 0.04
0.94	3.61–3.86	−6.18 ± 0.07	5.44–5.68	–6.07 ± 0.04	5.14–5.48	−5.93 ± 0.04	4.70–7.38	−5.78 ± 0.06
0.96	3.02–3.22	−7.20 ± 0.05	5.94–6.17	–7.06 ± 0.06	5.71–6.92	−6.92 ± 0.07	5.71–6.56	−6.75 ± 0.06
0.98	4.17–4.35	−8.55 ± 0.04	5.32–5.46	–8.35 ± 0.06	6.52–7.08	−8.14 ± 0.06	5.34–5.90	−7.94 ± 0.04
1.00	4.14–4.22	−10.01 ± 0.04	4.22–4.28	−9.64 ± 0.06	8.16–10.64	−9.26 ± 0.03	6.77–9.22	−8.89 ± 0.04

^a^ X_w_ is the mole fraction of water in a solvent mixture. ^b^
*m* is the concentration range of 1,4-dioxane for six to eight independent measurements. ^c^
± is the uncertainty.

**Table 5 ijms-24-08934-t005:** Standard molar enthalpy of solution ($\Delta_{\mathrm{sol}}H_{}^{o}$) and molality (*m*) of 12C4 in the NMF + W mixtures at *T* = (293.15, 298.15, 303.15, 308.15) K.

	*T* = 293.15 K		*T* = 298.15 K		*T* = 303.15 K		*T* = 308.15 K	
$x_{W}$ **^a^**	$\frac{m^{b}\cdot 10_{}^{3}}{\mathrm{mol}\cdot {\mathrm{kg}}^{-1}}$	$\frac{\Delta_{\mathrm{sol}}H_{}^{o}}{\mathrm{kJ}\cdot {\mathrm{mol}}^{-1}}$	$\frac{m^{b}\cdot 10_{}^{3}}{\mathrm{mol}\cdot {\mathrm{kg}}^{-1}}$	$\frac{\Delta_{\mathrm{sol}}H_{}^{o}}{\mathrm{kJ}\cdot {\mathrm{mol}}^{-1}}$	$\frac{m^{b}\cdot 10_{}^{3}}{\mathrm{mol}\cdot {\mathrm{kg}}^{-1}}$	$\frac{\Delta_{\mathrm{sol}}H_{}^{o}}{\mathrm{kJ}\cdot {\mathrm{mol}}^{-1}}$	$\frac{m^{b}\cdot 10_{}^{3}}{\mathrm{mol}\cdot {\mathrm{kg}}^{-1}}$	$\frac{\Delta_{\mathrm{sol}}H_{}^{o}}{\mathrm{kJ}\cdot {\mathrm{mol}}^{-1}}$
0.00	2.19–3.25	−7.54 ± 0.04 ^c^	2.17–2.32	−7.46 ± 0.06	2.52–4.07	−7.39 ± 0.04	2.97–3.04	−7.31 ± 0.05
0.10	3.02–3.47	−7.65 ± 0.04	3.18–3.35	−7.55 ± 0.07	3.95–4.79	−7.45 ± 0.04	2.94–3.35	−7.34 ± 0.06
0.20	2.44–3.69	−7.75 ± 0.05	2.14–2.40	−7.63 ± 0.05	2.39–3.14	−7.53 ± 0.06	2.97–3.64	−7.40 ± 0.06
0.30	3.48–4.71	−7.90 ± 0.06	2.95–3.19	−7.76 ± 0.05	2.40–3.14	−7.64 ± 0.06	3.02–3.59	−7.51 ± 0.07
0.40	5.98–8.15	−8.16 ± 0.06	2.11–2.23	−8.03 ± 0.07	2.96–2.99	−7.88 ± 0.08	3.23–4.03	−7.71 ± 0.07
0.50	5.20–5.66	−8.65 ± 0.06	2.37–2.74	−8.50 ± 0.05	3.40–4.61	−8.33 ± 0.08	4.26–4.73	−8.17 ± 0.08
0.60	4.58–4.94	−9.73 ± 0.08	3.64–3.76	−9.57 ± 0.08	3.20–5.14	−9.41 ± 0.07	3.28–3.69	−9.26 ± 0.08
0.70	4.65–5.58	−11.62 ± 0.04	3.31–3.53	−11.47 ± 0.06	3.87–4.26	−11.33 ± 0.05	3.67–4.42	−11.18 ± 0.04
0.80	3.32–4.07	−14.96 ± 0.06	3.81–3.92	−14.81 ± 0.08	4.87–5.20	−14.65 ± 0.04	3.30–3.72	−14.49 ± 0.05
0.90	1.56–3.14	−20.90 ± 0.06	2.63–2.79	−20.66 ± 0.05	4.27–4.94	−20.39 ± 0.05	3.47–4.02	−20.10 ± 0.04
0.92	2.50–3.62	−22.12 ± 0.06	2.54–2.62	−21.81 ± 0.06	2.44–4.05	−21.51 ± 0.06	3.74–3.81	−21.21 ± 0.05
0.94	2.45–3.26	−23.70 ± 0.04	3.13–3.42	−23.36 ± 0.06	2.69–2.98	−23.03 ± 0.04	3.28–3.90	−22.69 ± 0.06
0.96	2.70–3.06	−25.24 ± 0.04	2.96–3.39	−24.91 ± 0.04	2.14–3.55	−24.51 ± 0.05	3.93–4.03	−24.13 ± 0.07
0.98	1.98–2.82	−27.15 ± 0.04	3.04–3.08	−26.67 ± 0.04	2.68–2.99	−26.25 ± 0.05	3.17–3.52	−25.77 ± 0.07
1.00	1.71–2.77	−29.75 ± 0.04	5.33–5.24	−28.98 ± 0.04	2.65–5.99	−28.23 ± 0.04	1.97–4.15	−27.45 ± 0.03

^a^ X_w_ is the mole fraction of water in a solvent mixture. ^b^
*m* is the concentration range of 12C4 for six to eight independent measurements. ^c^
± is the uncertainty.

**Table 6 ijms-24-08934-t006:** Standard molar enthalpy of solution ($\Delta_{\mathrm{sol}}H_{}^{o}$) and molality (*m*) of 15C5 in the NMF + W mixtures at *T* = (293.15, 298.15, 303.15, 308.15) K.

	*T* = 293.15 K		*T* = 298.15 K		*T* = 303.15 K		*T* = 308.15 K	
$x_{W}$ **^a^**	$\frac{m^{b}\cdot 10_{}^{3}}{\mathrm{mol}\cdot {\mathrm{kg}}^{-1}}$	$\frac{\Delta_{\mathrm{sol}}H_{}^{o}}{\mathrm{kJ}\cdot {\mathrm{mol}}^{-1}}$	$\frac{m^{b}\cdot 10_{}^{3}}{\mathrm{mol}\cdot {\mathrm{kg}}^{-1}}$	$\frac{\Delta_{\mathrm{sol}}H_{}^{o}}{\mathrm{kJ}\cdot {\mathrm{mol}}^{-1}}$	$\frac{m^{b}\cdot 10_{}^{3}}{\mathrm{mol}\cdot {\mathrm{kg}}^{-1}}$	$\frac{\Delta_{\mathrm{sol}}H_{}^{o}}{\mathrm{kJ}\cdot {\mathrm{mol}}^{-1}}$	$\frac{m^{b}\cdot 10_{}^{3}}{\mathrm{mol}\cdot {\mathrm{kg}}^{-1}}$	$\frac{\Delta_{\mathrm{sol}}H_{}^{o}}{\mathrm{kJ}\cdot {\mathrm{mol}}^{-1}}$
0.00	2.92–3.24	−7.13 ± 0.04 ^c^	2.12–2.19	−6.73 ± 0.05	3.72–3.85	−6.33 ± 0.09	1.79–3.48	−5.96 ± 0.08
0.10	2.88–3.32	−7.75 ± 0.04	2.22–2.37	−7.35 ± 0.07	3.48–3.56	−6.96 ± 0.08	3.35–3.75	−6.57 ± 0.06
0.20	2.99–3.67	−8.17 ± 0.04	2.16–2.20	−7.78 ± 0.07	3.74–3.90	−7.38 ± 0.06	3.68–4.18	−6.99 ± 0.08
0.30	1.94–4.50	−8.98 ± 0.06	2.14–2.19	−8.57 ± 0.04	3.15–3.21	−8.19 ± 0.07	3.27–3.41	−7.81 ± 0.07
0.40	1.68–2.42	−9.61 ± 0.06	2.11–2.31	−9.22 ± 0.06	3.74–3.98	−8.82 ± 0.07	3.34–3.92	−8.43 ± 0.05
0.50	1.38–2.04	−10.87 ± 0.08	2.01–2.21	−10.47 ± 0.06	3.85–3.97	−10.09 ± 0.05	1.89–2.32	−9.71 ± 0.05
0.60	1.58–1.94	−13.00 ± 0.06	2.02–2.13	−12.62 ± 0.07	3.12–3.75	−12.23 ± 0.05	4.24–4.81	−11.82 ± 0.06
0.70	2.12–2.88	−16.56 ± 0.04	2.26–2.29	−16.19 ± 0.07	3.61–3.82	−15.76 ± 0.08	5.08–5.19	−15.36 ± 0.08
0.80	1.97–2.54	−22.20 ± 0.04	3.39–3.44	−21.75 ± 0.05	3.75–3.99	−21.34 ± 0.06	3.94–5.08	−20.90 ± 0.07
0.90	2.21–2.50	−29.42 ± 0.06	3.33–3.38	−28.86 ± 0.04	3.38–3.64	−28.26 ± 0.04	3.73–3.85	−27.70 ± 0.08
0.92	1.74–2.39	−31.50 ± 0.07	3.62–3.72	−30.84 ± 0.04	3.39–3.56	−30.12 ± 0.04	3.96–3.99	−29.50 ± 0.09
0.94	2.02–2.60	−33.45 ± 0.08	2.32–2.48	−32.53 ± 0.04	2.38–2.62	−31.62 ± 0.08	4.12–4.34	−30.75 ± 0.06
0.96	1.61–2.52	−35.80 ± 0.07	2.91–3.03	−34.68 ± 0.06	2.10–2.51	−33.55 ± 0.08	3.02–3.31	−32.54 ± 0.06
0.98	2.10–3.20	−38.23 ± 0.07	2.15–2.27	−36.98 ± 0.05	2.21–2.77	−35.72 ± 0.04	2.16–2.80	−34.54 ± 0.04
1.00	1.16–2.48	−41.24 ± 0.04	2.15–2.32	−39.80 ± 0.02	1.58–3.52	−38.37 ± 0.03	1.37–1.52	−37.03 ± 0.04

^a^ X_w_ is the mole fraction of water in a solvent mixture. ^b^
*m* is the concentration range of 15C5 for six to eight independent measurements. ^c^
± is the uncertainty.

**Table 7 ijms-24-08934-t007:** Standard molar enthalpy of solution ($\Delta_{\mathrm{sol}}H_{}^{o}$) and molality (*m*) of 18C6 in the NMF + W mixtures at *T* = (293.15, 298.15, 303.15, 308.15) K.

	*T* = 293.15 K		*T* = 298.15 K		*T* = 303.15 K		*T* = 308.15 K	
$x_{W}$ ^a^	$\frac{m^{b}\cdot 10_{}^{3}}{\mathrm{mol}\cdot {\mathrm{kg}}^{-1}}$	$\frac{\Delta_{\mathrm{sol}}H_{}^{o}}{\mathrm{kJ}\cdot {\mathrm{mol}}^{-1}}$	$\frac{m^{b}\cdot 10_{}^{3}}{\mathrm{mol}\cdot {\mathrm{kg}}^{-1}}$	$\frac{\Delta_{\mathrm{sol}}H_{}^{o}}{\mathrm{kJ}\cdot {\mathrm{mol}}^{-1}}$	$\frac{m^{b}\cdot 10_{}^{3}}{\mathrm{mol}\cdot {\mathrm{kg}}^{-1}}$	$\frac{\Delta_{\mathrm{sol}}H_{}^{o}}{\mathrm{kJ}\cdot {\mathrm{mol}}^{-1}}$	$\frac{m^{b}\cdot 10_{}^{3}}{\mathrm{mol}\cdot {\mathrm{kg}}^{-1}}$	$\frac{\Delta_{\mathrm{sol}}H_{}^{o}}{\mathrm{kJ}\cdot {\mathrm{mol}}^{-1}}$
0.00	0.62–0.83	9.27 ± 0.06 c	0.70–0.86	10.78 ± 0.04	2.40–3.17	12.32 ± 0.08	1.92–2.25	13.81 ± 0.06
0.10	0.72–1.74	8.62 ± 0.06	0.74–0.85	10.15 ± 0.06	2.64–2.82	11.62 ± 0.04	1.83–2.59	13.11 ± 0.06
0.20	2.11–2.27	7.99 ± 0.08	1.47–1.80	9.44 ± 0.06	2.96–3.24	12.95 ± 0.06	3.15–2.36	12.28 ± 0.08
0.30	2.14–2.68	7.34 ± 0.08	0.62–2.12	8.76 ± 0.05	1.05–1.61	10.18 ± 0.05	1.38–3.39	11.52 ± 0.06
0.40	2.80–3.30	6.53 ± 0.06	0.57–2.44	7.86 ± 0.05	1.28–1.88	9.22 ± 0.04	1.94–3.22	10.55 ± 0.07
0.50	1.87–2.51	5.49 ± 0.05	1.95–2.80	6.75 ± 0.06	1.72–2.10	8.03 ± 0.06	2.08–4.08	9.34 ± 0.05
0.60	2.08–2.17	3.66 ± 0.05	1.27–3.45	4.85 ± 0.08	3.24–3.69	6.08 ± 0.06	1.72–1.57	7.28 ± 0.06
0.70	1.50–1.85	0.50 ± 0.07	1.10–2.82	1.70 ± 0.07	3.36–3.43	2.91 ± 0.04	1.37–1.85	4.09 ± 0.08
0.80	1.29–1.90	−4.67 ± 0.08	1.50–1.89	−3.45 ± 0.08	3.81–3.93	−2.19 ± 0.04	1.68–2.17	−0.95 ± 0.05
0.90	0.76–1.46	−12.26 ± 0.07	0.52–0.76	−10.95 ± 0.05	1.23–1.96	−9.68 ± 0.06	3.49–4.16	−8.32 ± 0.06
0.92	1.02–1.12	−14.28 ± 0.04	1.25–1.65	−12.89 ± 0.05	2.27–3.17	−11.52 ± 0.08	2.15–3.14	−10.15 ± 0.08
0.94	0.94–1.01	−16.38 ± 0.04	1.32–1.38	−14.93 ± 0.04	2.51–3.09	−13.53 ± 0.08	1.98–2.57	−12.16 ± 0.04
0.96	0.61–0.71	−18.62 ± 0.04	1.17–1.38	−17.13 ± 0.06	2.49–2.63	−15.58 ± 0.04	1.77–2.51	−14.12 ± 0.06
0.98	0.24–0.60	−21.06 ± 0.04	1.22–1.47	−19.27 ± 0.04	1.42–1.54	−17.65 ± 0.04	1.66–2.81	−16.03 ± 0.04
1.00	5.43–6.52	−23.65 ± 0.02	1.92–2.44	−21.58 ± 0.03	2.55–5.87	−19.57 ± 0.02	2.48–4.64	−17.42 ± 0.02

^a^ X_w_ is the mole fraction of water in a solvent mixture. ^b^
*m* is the concentration range of 18C6 for six to eight independent measurements. ^c^
± is the uncertainty.

## Data Availability

Not applicable.

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
