# Peer review of "Effect of the Temperature on the Process of Preferential Solvation of 1,4-Dioxane, 12-Crown-4, 15-Crown-5 and 18-Crown-6 Ethers in the Mixture of N-Methylformamide with Water: Composition of the Solvation Shell of the Cyclic Ethers"

_ijms, 2023, doi:10.3390/ijms24108934_

Round 1
Reviewer 1 Report
In this work the heat of solution of 1,4-dioxane, 12-crown-4, 15-crown-5 and 18-crown-6 ethers in the mixture of N-methylformamide with water at four temperatures: 293.15 K, 298.15K, 303.15 K, 308.15 K were studied. This manuscript has several concerns such as interpretation of intermolecular interactions needs additional arguments.
“It is most likely related to the process of hydrophobic hydration of cyclic ethers”. The additional prove should be specified for this statement.
Lines 100-110 authors should be taken into account reorganization of hydrogen bond between NMF+W, and solvophobic effect of water.
“The studied cyclic ethers molecules are preferentially solvated by NMF molecules.” This conclusion should be briefly discussed from point of view intermolecular interaction in the manuscript text.
“The enthalpic effect of preferential solvation decreases with increasing temperature 286 for the cyclic ethers (1,4-dioxane, 12C4, 15C5 and 18C6)”. The additional interpretation in the manuscript text should be added.
Author Response
Dear Reviewer 1,
thank you very much for your very valuable and insightful comments. We tried to thoroughly answer all the questions and complete the suggested issues. Below are our responses to the issues raised in reviews. Moreover, English has been corrected by native speaker.
We hope that our additions are satisfactory and will allow publication of the manuscript.
“It is most likely related to the process of hydrophobic hydration of cyclic ethers”. The additional prove should be specified for this statement.
The following text has been added to our manuscript:
The analogous observation was discussed by Somsen [34] and Castagnolo [35] in the case of heat capacity of tetra-n-butyammonium bromide and by Desnoyers et al. [36] in the case of acetamide, dimethylsulfoxide, and acetone in mixed water-organic solvents. It was concluded that the shape of curves very similar to that obtained in our investigation is related with hydrophobic hydration of solute molecules.
Lines 100-110 authors should be taken into account reorganization of hydrogen bond between NMF+W, and solvophobic effect of water.
The following text has been added to our manuscript:
In the NMF + W mixture the change in the function is observed in a wider range of mixture composition than in the F + W mixture. Probably it is related to the structure of the NMF + W and F + W mixtures. The molecules of NMF forms a weaker network of hydrogen bonds than F [39]. The energy of hydrogen bonds in NMF is slightly lower than in F [40]. Formamide added to water destroys its original structure creating a new stable system [41,42]. In the case of pure NMF, molecules interact with each other through linear hydrogen bonds. In pure water, the molecules are also associated through hydrogen bonds. It turns out that the addition of NMF to water causes the formation of new hydrogen bonds between water molecules and NMF (C=O∙∙∙∙H–O), which are not much different from those found in water. This is indicated by heat capacity [43], viscosity or spectroscopic tests [44]. In addition, Assarsson and Eirich suggested that two water molecules should interact more strongly with the two lone pairs of electrons on the carbonyl oxygen, while the third molecule might interact less with the electron pair on the nitrogen atom. It can therefore be presumed that N-methylforamide can form three hydrogen bonds with water, whereas this type of interactions is less important in the F + W mixtures [45]. What's more, Marcus, while analyzing the preferential solvation parameter δx, noticed that the mixture of NMF + W behaves almost like an ideal one [9].
In general, the structure of the mixed solvent NMF + W is weaker than that of F + W, therefore there is a possibility of a specific interaction of 18C6 molecules with NMF molecules in a wider range of the mixture composition than in the case of the F + W mixture. On this basis, it can be assumed that 18C6 molecules form complexes with NMF molecules through hydrogen bonds. We can suppose that 18C6 molecules can form complexes by hydrogen bonds between hydrogen atoms of NMF molecule and electron pair of oxygen atoms of 18C6 molecule but also between hydrogen atoms of –CH3 group and electron pair of oxygen atoms of 18C6 molecule similarly to the hydrogen bonds of acetonitrile with oxygen atoms of 18C8 molecule [46] particularly in the medium and low water content in the mixture. In F + W mixture this type of interactions cannot occur.
“The studied cyclic ethers molecules are preferentially solvated by NMF molecules.” This conclusion should be briefly discussed from point of view intermolecular interaction in the manuscript text.
The following text has been added to our manuscript:
The equilibrium with the constant K described by Equation (5) represents the reverse process, i.e. the solvation by water molecules. Thus, in the case of preferential solvation by NMF molecules, the equilibrium constant of this process will be equal to K’ = 1/K (Table 1). As is seen in Table 1 the equilibrium constants of preferential solvation increase with increasing the ring size of cyclic ethers and decrease with increasing temperature. An increase in temperature causes a weakening of the interactions between molecules of water and NMF, as well as, between molecules of cyclic ethers and NMF due to an increase in thermal motion.
Specific interactions of cyclic ether molecules with NMF molecules may occur through hydrogen bonds between hydrogen atoms (one at the nitrogen atom and the other in the carbonyl group) with oxygen atoms in the cyclic ether molecules, especially in the area of medium and low water content in the mixture. In addition, molecules of 18C6 can form complexes with NMF molecules by interacting with the –CH3 group, which we wrote about in section 2.1.
“The enthalpic effect of preferential solvation decreases with increasing temperature 286 for the cyclic ethers (1,4-dioxane, 12C4, 15C5 and 18C6)”. The additional interpretation in the manuscript text should be added.
The following text has been added to our manuscript:
In the text:
As is seen in Figure 3 the exothermic, enthalpic effect of preferential solvation increases with increasing of cyclic ethers size and temperature. Comparing this effect for cyclic ethers, it can be seen that in the NMF + W mixture it is less negative than in the F + W mixture [16]. As the temperature increases, the interactions between the molecules of the mixed solvent start to weaken and it will be easier for one of the components of the mixture to interact with the molecules of the cyclic ethers.
In the conclusion:
The exothermic enthalpic effect of preferential solvation increases with increasing temperature for the cyclic ethers (1,4-dioxane, 12C4, 15C5 and 18C6).
Kind regards,
MaÅ‚gorzata Jóźwiak

Reviewer 2 Report
1. The abstract should be rewritten to emphasize the novelty of the work and to add accurate information about the main conclusions of this work.
2. Authors should expand the introduction to include recent research and provide enough information and explanation to make the article informative for a general audience.
3. The scientific and practical significance of this study should be emphasized more.
4. Add new references if possible. Of the 40 links, only 16 are from the last 20 years, of which only 6 are from the last 5 years.
5. The presentation of the Figures needs to be improved. It is necessary to increase the signatures and numbers on the axes. Figure 3 is best divided similarly to Figures 1 and 5.
Author Response
Dear Reviewer 2,
thank you very much for your very valuable and insightful comments. We tried to thoroughly answer all the questions and complete the suggested issues. Below are our responses to the issues raised in reviews. Moreover, English has been corrected by native speaker.
We hope that our additions are satisfactory and will allow publication of the manuscript.
- The abstract should be rewritten to emphasize the novelty of the work and to add accurate information about the main conclusions of this work.
The abstract has been changed and now there is as following:
The aim of the work was to analyze the preferential solvation process, and determine the composition of the solvation shell of cyclic ethers using the calorimetric method. The heat of solution of 1,4-dioxane, 12-crown-4, 15-crown-5 and 18-crown-6 ethers in the mixture of N-methylformamide with water was measured at four temperatures: 293.15 K, 298.15 K, 303.15 K, 308.15 K and the standard partial molar heat capacity of cyclic ethers has been discussed. 18-crown-6 (18C6) molecules can form complexes with NMF molecules through the hydrogen bonds between –CH3 group of NMF and the oxygen atoms of 18C6. Using the model of preferential solvation it was observed the cyclic ethers are preferentially solvated by NMF molecules. It has been proved that the molar fraction of NMF in the solvation shell of cyclic ethers is higher than that in the mixed solvent. The exothermic, enthalpic effect of preferential solvation of cyclic ethers increase with increasing the ring size and temperature. The increase in the negative effect of structural properties of the mixed solvent with increase of the ring size in the process of preferential solvation of the cyclic ethers indicates an increasing disturbance of the mixed solvent structure, which is reflected in the influence of the energetic properties of the mixed solvent.
- Authors should expand the introduction to include recent research and provide enough information and explanation to make the article informative for a general audience.
In Introduction the following text has been added:
The studies on preferential solvation are carried out for a long time, mostly using solubility measurements [6‒8]. Marcus extensively investigate this phenomenon utilizing the solubility of investigated substances in water [9], organic solvents [10] and in mixed solvents [11]. The aim of these studies is to observe whether the molecules undergo the self-association, preferentially interact with the co-solvent or the solute molecules are preferentially solvated by the molecules of the one of the components of mixed solvents. This is very important from the point of view of, e.g. pharmacology and effect of the drug or occurrence of chemical reactions or a change of the mechanism of the reaction due to change of the solvent. Similar studies are conducted using a spectroscopic method [12]. There are only a few publications on this topic that utilize the calorimetric method. Waghorn proposed the method to calculate the composition of the solvation shell using calorimetric method [13]. This method was used for the calculation [14] however, does not always give satisfactory results. Therefore, there are attempts to utilize another theory to analyze preferential solvation process. In the present publication, which is a continuation of our research [15,16], we used the model of preferential solvation proposed by Covington and modified by Balk and Somsen [17‒19] for the analysis of the calorimetric data from the point of view of calculation of the solvation shell of the solute in this process. For this purpose we chosen the cyclic ethers (1,4-dioxane, 12-crown-4 (12C4), 15-crown-5 (15C5) and 18-crown-6 (18C6)) and N‑methylformamide-water (NMF + W) mixed solvent.
- The scientific and practical significance of this study should be emphasized more.
In our opinion now, after revision the abstract and introduction contains sufficient information.
- Add new references if possible. Of the 40 links, only 16 are from the last 20 years, of which only 6 are from the last 5 years.
They have been added.
- The presentation of the Figures needs to be improved. It is necessary to increase the signatures and numbers on the axes. Figure 3 is best divided similarly to Figures 1 and 5.
It has been changed.
Kind regards,
MaÅ‚gorzata Jóźwiak

Reviewer 3 Report
The paper describes the effect of the cyclic ethers' size, the temperature, and the N-methylformamide-wather mixture on the standard molar solution enthalpy. The data were used to draw interesting conclusions related to the corona ether's behavior in this solvent mixture.
Points wothh of consideration:
L. 19 is it really calculated or proposed. If it was calculated than please show the model and schematic picture.
Simplicify: „This results an 21 increase in the positive energy content, which reflects the difficulties in the incorporation of cyclic 22 ether molecules into the enhanced structure of the mixed solvent.”
„is very important from the point of view of interactions between molecules of reactants” –do you mean between molecules and solvent?
L.42 solvents
L. 57 As can be seen
L. 89 Style: „This means that even if there are any changes in the structure of the solution, 89 they are small and not reflected in this function in this range of composition”
L.102 Please, be consequent, use all the time ‘the’ in front of abbreviations like 18C6 and so on.
L.128 you mean “disolved substancje”?
L.184 red coma
L.149 this is not equation but reaction!
L.169-174 Do you mean hydrogen bonds interactions or with water or NMF? Please clarify!
L.175 Increase of temperature almost always increase solubility.
L.239 “the solvent structure is enhanced” is is true?
Table 3, the 1 an4 columns can be a bit extender. It will be mor easy to read.
L. 261 „was not 261 noticed” maybe was not reported In literature?
L. 279 investigation , L. 281 leads
L. 284 –red coma
L. 288 „it is better: „The total number of NMF and W molecules in the solvation sphere of the molecules of cyclic ethers increases with the increasing ring size of the ethers. ”
L.292 it is not clear and [potentially can be wrong conclusion.
A number of small grammar or style mistakes. Please, refer to the points raised above.
Author Response
Dear Reviewer 3,
thank you very much for your very valuable and insightful comments. We tried to thoroughly answer all the questions and complete the suggested issues. Below are our responses to the issues raised in reviews. Moreover, English has been corrected by native speaker.
We hope that our additions are satisfactory and will allow publication of the manuscript.
- 19 is it really calculated or proposed. If it was calculated than please show the model and schematic picture.
Simplicify: „This results an 21 increase in the positive energy content, which reflects the difficulties in the incorporation of cyclic 22 ether molecules into the enhanced structure of the mixed solvent.”
The abstract has been changed and now there is as following:
The aim of the work was to analyze the preferential solvation process, and determine the composition of the solvation shell of cyclic ethers using the calorimetric method. The heat of solution of 1,4-dioxane, 12-crown-4, 15-crown-5 and 18-crown-6 ethers in the mixture of N-methylformamide with water was measured at four temperatures: 293.15 K, 298.15 K, 303.15 K, 308.15 K and the standard partial molar heat capacity of cyclic ethers has been discussed. 18-crown-6 (18C6) molecules can form complexes with NMF molecules through the hydrogen bonds between –CH3 group of NMF and the oxygen atoms of 18C6. Using the model of preferential solvation it was observed the cyclic ethers are preferentially solvated by NMF molecules. It has been proved that the molar fraction of NMF in the solvation shell of cyclic ethers is higher than that in the mixed solvent. The exothermic, enthalpic effect of preferential solvation of cyclic ethers increase with increasing the ring size and temperature. The increase in the negative effect of structural properties of the mixed solvent with increase of the ring size in the process of preferential solvation of the cyclic ethers indicates an increasing disturbance of the mixed solvent structure, which is reflected in the influence of the energetic properties of the mixed solvent.
„is very important from the point of view of interactions between molecules of reactants” –do you mean between molecules and solvent?
Now in the text there is:
Not only pure solvents are taken into account, but also mixed water-organic and organic-organic mixtures that significantly affect the solvation of the reactants. The solvation process plays a very important role in the interactions between molecules of solvent, solvent and solute, as well as solute and solute [1‒3].
L.42 solvents
It has been changed.
- 57 As can be seen.
It has been changed.
- 89 Style: „This means that even if there are any changes in the structure of the solution, 89 they are small and not reflected in this function in this range of composition”
It has been changed and now there is:
This means that even if there are any changes in the structure of the solution, they are small and do not influence the shape of function in this range of composition.
L.102 Please, be consequent, use all the time ‘the’ in front of abbreviations like 18C6 and so on.
It has been corrected.
L.128 you mean “disolved substancje”?
Now there is :
Solute molecules
L.184 red coma
It has been corrected.
L.149 this is not equation but reaction!
It has been corrected.
L.169-174 Do you mean hydrogen bonds interactions or with water or NMF? Please clarify!
L.175 Increase of temperature almost always increase solubility.
Of course, it depends on the enthalpy of solution (whether the process is endo- or exothermic). In the case of interactions and movement of the molecules an increase in temperature causes a weakening of the interactions between molecules in the system due to an increase in thermal motion.
This text has been changed and now there is as following:
The equilibrium with the constant K described by Equation (5) represents the reverse process, i.e. the solvation by water molecules. Thus, in the case of preferential solvation by NMF molecules, the equilibrium constant of this process will be equal to K’ = 1/K (Table 1). As is seen in Table 1 the equilibrium constants of preferential solvation increase with increasing the ring size of cyclic ethers and decrease with increasing temperature. An increase in temperature causes a weakening of the interactions between molecules of water and NMF, as well as, between molecules of cyclic ethers and NMF due to an increase in thermal motion.
Specific interactions of cyclic ether molecules with NMF molecules may occur through hydrogen bonds between hydrogen atoms (one at the nitrogen atom and the other in the carbonyl group) with oxygen atoms in the cyclic ether molecules, especially in the area of medium and low water content in the mixture. In addition, molecules of 18C6 can form complexes with NMF molecules by interacting with the –CH3 group, which we wrote about in section 2.1.
L.239 “the solvent structure is enhanced” is is true?
This text has been changed and now there is as following:
In our case, NMF interacts with water through hydrogen bonds and forms a new, also stable structure with water, which is reflected also in the negative contribution of the structural properties and the positive contribution of the energetic properties of the mixed solvent to the solvation of the cyclic ethers.
Table 3, the 1 an4 columns can be a bit extender. It will be more easy to read.
It has been changed.
- 261 „was not 261 noticed” maybe was not reported In literature?
We have not observed the concentration dependence of the enthalpy of dissolution of cyclic ethers in our study.
- 279 investigation,
It has been changed.
- 281 leads
It has been changed.
- 284 –red coma
It has been changed.
- 288 „it is better: „The total number of NMF and W molecules in the solvation sphere of the molecules of cyclic ethers increases with the increasing ring size of the ethers. ”
It has been changed.
L.292 it is not clear and [potentially can be wrong conclusion.
This text has been changed and now there is:
The increase in the negative effect of the mixed solvent structural properties with the increase in the size of the cyclic ring in the process of preferential solvation of the cyclic ethers indicates an increasing disturbance of the mixed solvent structure, which is reflected in the influence of the energetic properties of the mixed solvent.
Kind regards,
MaÅ‚gorzata Jóźwiak